

# Correlation between gross motor coordination and basic coordination capacities in normal-weight and overweight/obese children aged 9–10 years

Yuan Sui[1], Lin Cui[2], Binbin Jia[3], Xiangyang Ding[1], Min He[1], Yingen Da[1], Yue Shi[1], Fei Li[1,4] and Pan Li[1,4]

[1] School of Athletic Performance, Shanghai University of Sport, Shanghai, China
[2] Department of Military Physical Education, National University of Defense Technology, Beijing, China
[3] School of Sports Training, Wuhan Sports University, Wuhan, China
[4] Shanghai Key Lab of Human Performance, Shanghai University of Sport, Shanghai, China

Corresponding authors
Fei Li, lifei@sus.edu.cn
Pan Li, lipan@sus.edu.cn

## ABSTRACT

**Background:** Gross motor coordination (GMC) plays a crucial factor in children's motor development and daily activities. It encompasses various sub-capacities, such as spatial orientation, rhythm, and motor reaction, collectively referred to as basic coordination capacities (BCC). However, children who are overweight and obese (OW/OB) often display poorer GMC. This study aims to examine the impact of gender and weight status (BMI categories) on children's GMC and BCC. It also seeks to investigate the impact of BCC and BMI on GMC.

**Method:** The study involved 266 participants, 135 in the NW group (boys: $n = 75$; girls: $n = 60$) and 131 in the OW/OB group (boys: $n = 68$; girls: $n = 63$). An NW status is defined by a BMI z-score between $\geq -2SD$ to $\leq 1SD$, while an OW/OB status corresponds to a BMI z-score $> 1SD$. Physical activity was assessed using the Physical Activity Questionnaire for Children, developed by the University of Saskatchewan, Canada. We used six field tests to evaluate BCC, including single leg standing test (static balance), YBT (dynamic balance), rhythmic sprint test (rhythm), reaction time test (motor reaction), target standing broad test (kinesthetic differentiation), and numbered medicine ball running test (spatial orientation). GMC was evaluated with Kiphard-Schilling's Body Coordination Test (KTK).

**Result:** The motor quotient (MQ) was primarily affected by weight status (F = 516.599, $p < 0.001$; gender: F = 6.694, $p = 0.01$), with no significant interaction effect (F = 0.062, $p = 0.803$). In BCC, gender had a significant main effect on rhythm capacity (F = 29.611, $p < 0.001$) and static balance (F = 11.257, $p = 0.001$) but did not significant influence other sub-capacities ($p > 0.05$). Weight status impacted dynamic balance (F = 11.164, $p = 0.001$). The interaction of gender and weight status significantly impacted motor reaction (F = 1.471, $p = 0.024$) and kinesthetic differentiation (F = 5.454, $p = 0.02$), but did not affect other sub-capacities ($p > 0.05$). The physical activity was not significant affected by gender (F = 0.099, $p = 0.753$), weight status (F = 0.171, $p = 0.679$) and the interactions of two variables (F = 0.06, $p = 0.806$). In the regression analysis, except motor reaction ($p > 0.05$), other BCC sub-capacities influenced GMC to varying extents ($\beta = -0.103$–$0.189$, $p < 0.05$).

Nonetheless, only two types of balance significantly mediated the relationship between BMI and GMC (BMI→MQ: $\beta = -0.543$, $p < 0.001$; BMI→YBT: $\beta = -0.315$, $p < 0.001$; BMI→SLS: $\beta = -0.282$, $p < 0.001$; SLS→MQ: $\beta = 0.189$, $p < 0.001$; YBT→MQ: $\beta = 0.182$, $p < 0.001$).

**Conclusion:** Compared to gender, the main effect of weight status on most GMC and BCC's sub-capacities was more pronounced. OW/OB children exhibited poorer GMC, which is related to their reduced static and dynamic balance due to excess weight. Kinesthetic differentiation, spatial orientation, and rhythm capacity are not significantly associated with BMI, but these sub-capacities positively influence gross motor coordination (GMC), except for hand-eye motor reaction.

# INTRODUCTION

"Coordination" refers to the efficient and smooth motor performance during human movement. It applies to any voluntary motor action and is crucial for the development of children's motor skills, behaviors and competence (*Haga, 2009*; *Hulteen et al., 2018*). Depending on the type of motor action, coordination is categorized into general motor coordination, specific motor coordination, segmental motor coordination, and others (*Gentier et al., 2013*; *Yin et al., 2023*). Among these, gross motor coordination (GMC) involves actions that use two or more parts, or the whole body, reflecting a complex interplay of neurological and neuromotor processes (*Giuriato et al., 2021*). This foundational coordination should be developed early in childhood. Research suggested that good GMC is essential for children's successful participation in daily life and sports activities. It also fosters a healthy and optimistic attitude towards life in adulthood (*Wade, 1986*; *Haga, 2009*; *Reyes et al., 2019*). On the other hand, children with poor GMC at an early age may have lower perceived motor competence, physical health, and engagement in physical activity throughout their lifespan, potentially leading to an unhealthy physical appearance, such as overweight or obesity (OW/OB). Several studies have confirmed a significant negative correlation between GMC and physical activity, cardiorespiratory fitness, and weight status (*Stodden et al., 2008*; *Chagas & Batista, 2015*; *Burton et al., 2023*). Additionally, a positive correlation between GMC and academic performance (*Fernandes et al., 2016*; *Rosa Guillamón, García Cantó & Martínez García, 2020*). Obviously, GMC is the center element for child's long-term development pattern.

However, a 2022 national survey ($n = 193{,}997$) indicated that the prevalence of OW/OB among children in different provinces of China is around 20–25%, ranking it high globally (*Abarca-Gómez et al., 2017*). This further suggests an increased likelihood of a deficit in GMC among Chinese children, both now and potentially in the future. Several studies have identified OW/OB as a reliable negative predictor of GMC (*D'Hondt et al., 2014*; *Lopes et al., 2018*). However, most existing research is based on samples from North America and Australia, overlooking the fact that geographic regions, cultures, and social environments

influence children's GMC development (*Venetsanou & Kambas, 2010*; *Pereira et al., 2021*). Thus, it is imperative to research and understand the GMC among Chinese children. Additional studies from China and other Asian regions are necessary to establish the potential association between weight status and GMC.

Not only that, recent findings suggested that poor GMC is increasingly prevalent among school-aged children. *Hardy et al. (2012)* in a survey of 8,000 Australian school students, observed that many primary school students struggle with fundamental movement skills such as kicking and throwing. This issue can be attributed to the campus-based lifestyle, which often promotes low levels of physical activity and prevalent sedentary behaviors among children. While it is widely recognized that increasing physical activity levels could be a viable means to improve children's GMC, consensus on the most effective type and method of physical activity remains elusive. Therefore, the study aims to explore the capacities related to GMC, providing a basis for development strategies.

Children's GMC is influenced by internal and external factors. Internal factors are regarded as the relatively stable personal capacities related to GMC, which are formed through long-term life experiences. Researchers, based on a structural model of motor ability and considering children's growth and development, have identified five essential and early-developing capacities, termed "basic coordination capacities (BCC)", including balance (static and dynamic balance), rhythm, kinesthetic differentiation, spatial orientation and motor reaction. These sub-capacities largely reflect and determine the motor control and execution. Due to the complexity and broad scope of motor coordination, as well as uncertainty in understand, the comprehension of its structure remains quite vague. Additionally, a significant negative correlation between children OW/OB and GMC has been confirmed in numerous studies, but there is a lack of research exploring the reasons from the perspective of individual capacities (*Šimonek, 2014*).

Reflecting on the above, this study has two main objectives: 1) to examine the impact of gender and weight status (BMI categories) on children's GMC and BCC, and 2) to explore the relationship between BMI, GMC, and BCC. The results of this study may help identify the individual capacities contributing to the negative association between weight status and GMC. In turn, this aids in laying the foundation for refining the children's GMC model and provides an objective basis for developing measures for teachers and coaches.

# MATERIALS AND METHODS

## Participants and design

A sample size calculation was performed using G*power 3.1 software to ensure sufficient statistical power for this study. We assumed an effect size of 0.5, an alpha level of 0.05, and a power of 0.95. The results suggested that a minimum total sample size of 210 participants (N1 = N2 = 105) is required.

Participants were divided into two groups based on their BMI z-scores. The World Health Organization (WHO) growth standards for children defined NW status as a BMI z-score between $\geq -2SD$ to $\leq 1SD$, while OW/OB status was determined by a BMI z-score $> 1SD$ (*de Onis et al., 2007*). Some students were unable to fully participate due to such health or academic reasons. A total of 266 participants were ultimately included in the

study, with 135 in the NW group (boys: $n = 75$; girls: $n = 60$) and 131 in the OW/OB group (boys: $n = 68$; girls: $n = 63$).

The participant inclusion criteria were as follows: 1) age between 9 and 10 years; 2) no injuries in the past 3 months; and 3) completion of all test procedures. The exclusion criteria comprised: 1) any diseases, such as intellectual disabilities, muscular dystrophy, heart disease, other skeletal muscle diseases, and organic diseases; 2) weight status categorized as thin or emaciated.

WRITTEN informed consent were obtained from their parents. The Ethics Committee of the Shanghai University of Sport (Approval Number: 102772023RT108) granted ethical approval on October 13, 2023. All student participation was voluntary, and they could withdraw from the study at any time during the testing period if they experienced any discomfort.

## Measurement

A team of skilled researchers conducted evaluations on anthropometry, gross motor coordination, and five basic coordination sub-capacities, following established protocols. These assessments were carried out in August 2023 in the gymnasiums and playgrounds of selected schools. To minimize any proficiency-related effects on the results, participants received thorough instructions and demonstrations from the researchers and teachers, including 2–3 practice trials before the actual testing. Before the test began, researchers organized about 15 min of warm-up activities for participants, including jogging, core activation, and dynamic stretching.

## Physical activity assessments

The physical activity assessments in this study used the Physical Activity Questionnaire for Children (PAQ-C) developed by the University of Saskatchewan, Canada. The PAQ-C has shown good reliability and validity in tests with Chinese children, with a Cronbach's alpha of 0.79, composite reliability ($\rho$) of 0.81, and intraclass correlation coefficient (ICC) of 0.82. The questionnaire includes 10 items scored on a 1–5 scale. The total score is the sum of the item scores, with the PAQ-C score being the average of items 1 to 9 (item 10 is excluded as it assesses if illness or events prevented participation in activities). Scores below 2.33 indicate low activity, 2.33–3.66 indicate moderate activity, and above 3.66 indicate high activity (*Su et al., 2014*). The PAQ-C was completed in the classroom under the supervision of teachers and researchers, with a research assistant explaining the requirements and clarifying any questions.

## Anthropometry assessments

The school authorities helped acquire BMI data by providing results from a centralized physical fitness assessment conducted 1 month prior to this study. Additionally, we calculated BMI z-scores using Flemish growth references. These scores act as a substitute for adiposity, considering age and gender, and quantify a child's BMI deviation from the mean of the reference population in standard deviation units.

## Gross motor coordination assessments

The Kiphard-Schilling's body coordination test battery (KTK) was used to assess GMC. This test series, developed by Ernst J. Kiphard and Gerhard Schilling in the 1970s (*Kiphard & Schilling, 1974*), is designed for children aged 5 to 14. The KTK typically includes the following tests:

Walking backward (WB): Subjects are asked to walk backward three times on three beams of varying widths (6, 4.5, and 3 cm). All beams are three meters long. The total steps taken before falling from the beam are counted for each trial. The maximum score for each trial is eight steps.

Hopping for height (HH): Subjects perform one-legged jumps over progressively higher obstacles (a foam square measuring 50 cm × 20 cm × 5 cm). After a successful hop on each foot, the height is increased by adding a square.

Moving sideways (MS): Subjects move as many boards as possible within a specified time. Starting with both feet on one platform (25 cm × 25 cm × 2 cm supported on four legs 3.7 cm high) and holding an identical platform in their hands, they step onto the second platform. Two points are given for each successful transfer from one platform to the other. The total points in 20 s are counted and summed over two trials.

Jumping sideways (JS): Subjects are asked to continuously jump sideways between obstacles (a small beam) with both legs together as fast as possible for 15 s in a rectangular area (50 cm × 100 cm). The total number of correct jumps from the two trials is added together as the final raw result.

Motor quotient (MQ): The raw results for each test item are converted to a standardized score relative to sex- and age-specific reference values for the population upon which KTK was established. The sum of these standardized scores provides an overall motor quotient (MQ) (*Kiphard & Schilling, 1974*).

## Basic coordination capacities assessments

Six tests were selected from 25 existing ones to evaluate BCC. The inclusion criteria for these testing methods were as follows: 1) They have good reliability and validity; 2) they can be conducted in a field setting; 3) considering that children in this age group have not fully developed cognitive and motor systems, the chosen tests are easy for them to understand and perform; 4) they focus on body-weight and gross motor behaviors.

Balance: The single-leg standing test (SLS) was used to measure static balance (*Condon & Cremin, 2014*). Participants were asked to stand on one leg, hands on hips, with the non-supporting leg raised 10 to 20 cm off the ground. The trial ended when the participant's torso swayed significantly, they moved their foot for support, or the non-supporting leg touched the ground. Each leg was tested three times, with the best performance timed to an accuracy of 0.01 s recorded. The Y-Balance Test (YBT) was used to measure dynamic balance (*Fusco et al., 2020*). The participant's leg length was initially measured to the nearest 0.5 cm while they were lying down, wearing flip-flops, from the anterior superior iliac spine to the most distal point of the medial ankle. During the test, the participant stood on one foot on the center plate and extended and pushed the reach

indicator along the Y-axis in three directions. The best of three trials for each direction was recorded to an accuracy of 0.5 cm. The result was expressed as the average reach distance in the three directions relative to the leg length, multiplied by 100%. The mean value of both legs' results was calculated for the final score.

Kinesthetic differentiation: The target standing broad jump (TSBJ) was used to evaluate kinesthetic differentiation (Đolo, Grgantov & Milić, 2019). Participants performed three broad jumps. The farthest distance was marked as 75% of the target. Participants then tried to jump and land their heels as close to this mark as possible. The variance between the three jumps was recorded. The mean difference was calculated and used as the outcome. A smaller discrepancy indicates superior kinesthetic differentiation.

Motor reaction: Reaction time test (RT) (electronic reaction time testing device "FYS-II": Zhejiang Psychological Instrument Company, Hangzhou, China) was used to gauge motor reaction (Chang-sheng, Cheng-mou & Pei-su, 2006). Subjects placed their hand over the device and were instructed to press the button quickly when the photoelectric signal activated. This procedure was repeated three times, with the fastest reaction time recorded accurately to 0.001 s.

Rhythm capacity: Rhythm capacity was assessed through the rhythmic sprint test (RS) (Alper & Mustafa, 2018). As delineated in Fig. 1, participants were required to complete two 30-m sprints followed by a tempo run, with sprint times being recorded by photogates (Fusion Sport, Australia). For the tempo run, participants aimed to step into a specified circle of 11 while at a full sprint. The final score was determined by subtracting the tempo run time from the 30-m sprint time, accurate to 0.01 s. A shorter duration suggested a better rhythm ability.

Spatial orientation: It was assessed using the numbered medicine ball running test (NMBR) (Alper & Mustafa, 2018). As illustrated in Fig. 2, participants started from the "START" point and completed six runs in total. The first three runs were directed to specific markers based on the tester's numerical commands. After a 3-min break, participants completed three more runs without guidance (each run to a different marker bucket). The best performance was recorded and the difference was calculated, accurate to 0.01 s. Shorter times indicate superior spatial localization ability.

## Data collection

Data were collected from September 2023 to October 2023. Each student took approximately 40 to 60 min to complete all tests. However, due to exclusion criteria and lack of informed parental consent, the results of 34 children were excluded. As a result, the sample size was 266. This included 135 in the NW group (boys: $n = 75$, girls: $n = 60$) and 131 in the OW/OB group (boys: $n = 68$, girls: $n = 63$).

## Statistical analyses

The Shapiro-Wilk test, P-P plot, and Q-Q plot were used to analyze the distribution of data. Consequently, continuous variables were presented as mean (standard deviation (SD)). To evaluate the effects of gender and weight status on PA, BMI, GMC and BCC, we conducted a two-way ANOVA. This method allowed us to assess the main effects and

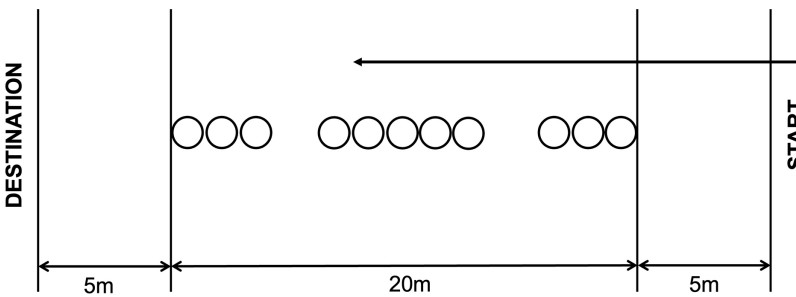

**Figure 1  Rhythmic sprint test.**               

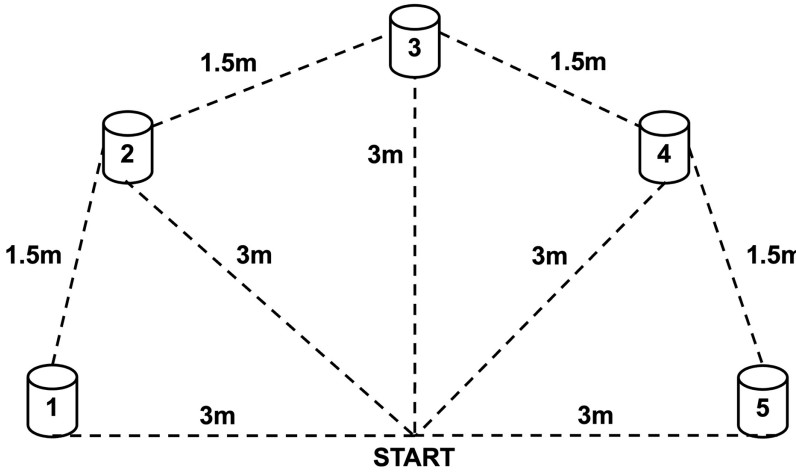

**Figure 2  Numbered medicine running test.**     

interaction effect between the two independent variables: gender (male and female) and weight status (NW and OW/OB). The associations between PA, GMC, BCC and BMI were examined using Pearson's correlation coefficients, with correlations deemed "small" for $r = 0.1–0.3$, "moderate" for $r = 0.3–0.5$, "large" for $r = 0.5–0.7$, "very large" for $r = 0.7–0.9$, and "extremely large" for $r = 0.9–1.0$ (*Hopkins et al., 2009*). To examine mediating effects, a bootstrapping technique with the PROCESS macro was applied. BMI was treated as the independent variable, and MQ was the dependent variable. BCC—static balance, dynamic balance, Kinesthetic differentiation, spatial orientation, and rhythm ability—as mediating variables. The analysis involved 5,000 bootstrap samples to generate percentile-based confidence intervals, with a 95% confidence level applied. Data preparation and analysis were conducted using the IBM SPSS Statistics version 27.0 (IBM Corp. Armonk, NK, USA) and PROCESS version 4.2 for SPSS. A *p*-value of less than 0.05 was established as the threshold for statistical significance in this study.

## RESULTS

The results showed that weight status ($F = 516.599$, $p < 0.001$) and its interaction with gender ($F = 6.694$, $p = 0.01$) significantly affected children's BMI, but gender did not have a significant effect ($F = 1.511$, $p = 0.22$). The gender ($F = 0.099$, $p = 0.753$) and weight status ($F = 0.171$, $p = 0.679$), as well as the interaction between the two ($F = 0.06$, $p = 0.806$), had

**Table 1 The effects of gender and weight status on PA, BMI, GMC and BCC.**

| Index | | NW | OW/OB | Total | G | | WS | | G*WS | |
|---|---|---|---|---|---|---|---|---|---|---|
| | | | | | F | p | F | p | F | p |
| **Physical activity** | Boys | 2.80 (0.88) | 2.81 (0.75) | 2.81 (0.80) | | | | | | |
| | Girls | 2.74 (0.74) | 2.81 (0.62) | 2.76 (0.70) | 0.099 | 0.753 | 0.171 | 0.679 | 0.06 | 0.806 |
| | Total | 2.77 (0.79) | 2.81 (0.71) | 2.79 (0.75) | | | | | | |
| **Body mass index** | Boys | 16.04 (0.82) | 23.24 (3.01) | 20.67 (4.25) | | | | | | |
| | Girls | 16.43 (1.01) | 22.16 (2.89) | 18.28 (3.26) | 1.511 | 0.22 | 516.599 | <0.001 | 6.694 | 0.01 |
| | Total | 16.28 (0.96) | 22.20 (2.95) | 19.54 (3.97) | | | | | | |
| **Gross motor coordination** | | | | | | | | | | |
| **Walking backward** | Boys | 82.20 (15.07) | 68.34 (13.81) | 73.29 (15.70) | | | | | | |
| | Girls | 86.96 (17.63) | 69.85 (13.48) | 81.40 (18.22) | 2.494 | 0.116 | 60.747 | <0.001 | 0.671 | 0.413 |
| | Total | 85.20 (16.83) | 68.82 (13.67) | 77.13 (17.38) | | | | | | |
| **Hopping for height** | Boys | 89.18 (9.67) | 76.62 (12.84) | 81.11 (13.23) | | | | | | |
| | Girls | 73.66 (11.15) | 73.66 (12.37) | 80.78 (12.76) | 7.924 | 0.005 | 56.222 | <0.001 | 0.715 | 0.398 |
| | Total | 85.71 (11.12) | 75.69 (12.37) | 80.78 (12.76) | | | | | | |
| **Jumping sideways** | Boys | 113.76 (13.39) | 103.51 (14.97) | 107.17 (15.20) | | | | | | |
| | Girls | 107.56 (12.37) | 97.26 (15.97) | 104.22 (14.41) | 11.589 | 0.001 | 31.699 | <0.001 | 0 | 0.995 |
| | Total | 109.86 (13.06) | 101.56 (15.5) | 105.77 (14.88) | | | | | | |
| **Moving sideways** | Boys | 86.38 (11.97) | 77.61 (11.50) | 80.74 (12.37) | | | | | | |
| | Girls | 81.95 (10.00) | 76.41 (10.75) | 80.15 (10.53) | 3.871 | 0.005 | 25.059 | <0.001 | 1.278 | 0.259 |
| | Total | 83.59 (10.94) | 77.24 (11.24) | 80.46 (11.52) | | | | | | |
| **Motor quotient** | Boys | 92.88 (8.82) | 81.52 (10.46) | 85.58 (11.28) | | | | | | |
| | Girls | 90.04 (9.26) | 79.31 (9.57) | 86.55 (10.60) | 3.083 | 0.044 | 77.844 | <0.001 | 0.062 | 0.803 |
| | Total | 91.10 (9.17) | 80.82 (10.21) | 86.04 (10.96) | | | | | | |
| **Basic coordination capacities** | | | | | | | | | | |
| **Static balance** | Boys | 40.83 (18.24) | 35.11 (18.62) | 37.16 (18.62) | | | | | | |
| | Girls | 49.58 (15.24) | 41.52 (17.87) | 46.94 (16.51) | 11.257 | 0.001 | 9.3 | 0.003 | 0.268 | 0.605 |
| | Total | 46.34 (16.86) | 37.12 (18.62) | 41.80 (18.29) | | | | | | |
| **Dynamic balance** | Boys | 91.29 (8.81) | 86.13 (9.23) | 87.97 (9.38) | | | | | | |
| | Girls | 90.85 (13.74) | 86.75 (8.11) | 89.52 (12.32) | 0.004 | 0.947 | 11.164 | 0.001 | 0.15 | 0.699 |
| | Total | 91.01 (12.12) | 86.32 (8.87) | 88.70 (10.88) | | | | | | |
| **Spatial orientation** | Boys | 2.73 (1.80) | 2.63 (1.73) | 2.67 (1.75) | | | | | | |
| | Girls | 2.30 (1.49) | 2.44 (1.49) | 2.34 (1.49) | 2.227 | 0.137 | 0.01 | 0.921 | 0.399 | 0.561 |
| | Total | 2.49 (0.58) | 2.32 (0.60) | 2.51 (1.63) | | | | | | |
| **Kinesthetic differentiation** | Boys | 7.24 (3.73) | 6.74 (3.12) | 6.92 (3.55) | | | | | | |
| | Girls | 6.52 (3.55) | 8.32 (5.14) | 7.11 (4.31) | 0.765 | 0.383 | 1.72 | 0.191 | 5.454 | 0.02 |
| | Total | 6.79 (3.62) | 7.23 (4.03) | 7.01 (3.83) | | | | | | |
| **Rhythm capacity** | Boys | 2.26 (0.54) | 2.17 (0.55) | 2.20 (0.54) | | | | | | |
| | Girls | 2.56 (0.56) | 2. 64 (0.58) | 2.59 (0.56) | 29.611 | <0.001 | 0.003 | 0.959 | 1.471 | 0.226 |
| | Total | 2.46 (1.62) | 2.57 (1.65) | 2.38 (0.59) | | | | | | |

| Index | | NW | OW/OB | Total | G | | WS | | G*WS | |
|---|---|---|---|---|---|---|---|---|---|---|
| | | | | | F | p | F | p | F | p |
| Motor reaction | Boys | 0.49 (0.06) | 0.50 (0.05) | 0.50 (0.06) | | | | | | |
| | Girls | 0.51 (0.05) | 0.48 (0.05) | 0.50 (0.05) | 0.011 | 0.916 | 1.772 | 0.184 | 1.471 | 0.024 |
| | Total | 0.500 (0.054) | 0.493 (0.052) | 0.50 (0.05) | | | | | | |

**Note:**
Normal-weight group, NW; Overweight and obese, OW/OB; Gender, G; Weight status, WS; the interaction of weight status and gender, G*WS. All data are presented using "mean (standard deviation)".

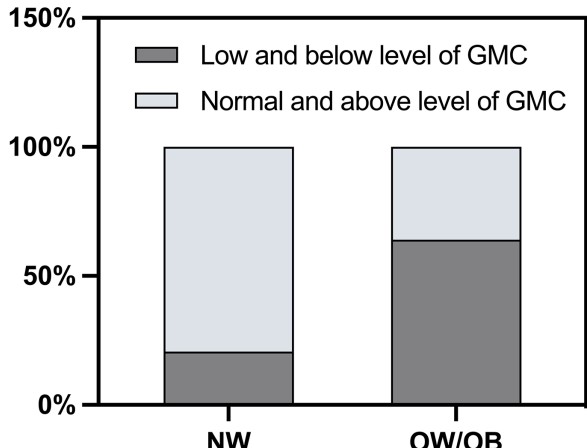

**Figure 3** The distribution of different GMC levels in NW and OW/OB children.

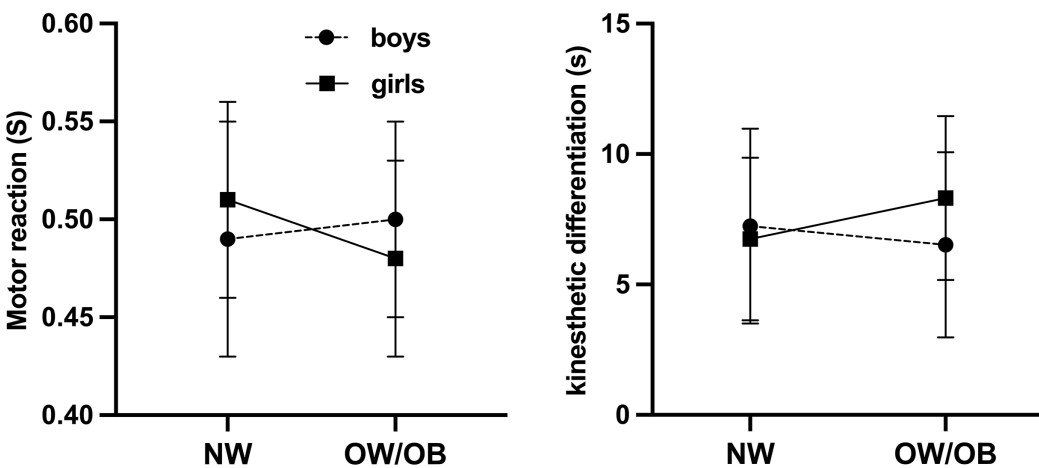

**Figure 4** The interaction between gender and weight status on motor reaction and kinesthetic differentiation.

**Table 2 Associations between BMI, PA, GMC and BCC.**

|  | Mean (SD) | 1 | 2 | 3 | 4 | 5 | 6 | 7 | 8 | 9 |
|---|---|---|---|---|---|---|---|---|---|---|
| 1. BMI | 19.64 (3.99) | – | | | | | | | | |
| 2. MQ | 86.04 (10.96) | −0.543** | – | | | | | | | |
| 3. TSBJ | 7.01 (3.83) | 0.014 | −0.156* | – | | | | | | |
| 4. NMBR | 2.51 (1.63) | 0.024 | −0.159** | 0.008 | – | | | | | |
| 5. SLS | 41.80 (18.29) | −0.282** | 0.366** | −0.127* | 0.038 | – | | | | |
| 6. YBT | 88.70 (10.88) | −0.315** | 0.402** | 0.003 | −0.151* | 0.233** | – | | | |
| 7. RT | 0.497 (0.535) | −0.058 | −0.056 | 0.063 | 0.162** | −0.052 | −0.03 | – | | |
| 8. RS | 2.38 (0.59) | −0.074 | −0.172** | 0.136* | 0.055 | −0.009 | −0.129* | 0.190** | – | |
| 9. PA | 2.29 (0.75) | 0.065 | −0.012 | −0.053 | −0.05 | −0.114 | −0.063 | −0.083 | 0.034 | – |

Notes:
* $p < 0.05$.
** $p < 0.01$.
Standard deviation, SD; Body mass index, BMI; motor quotient, MQ; Target standing broad jump, TSBJ; Numbered medicine ball running test, NMBR; single leg standing test, SLS; Y-balance test, YBT; Reaction time test, RT; Rhythmic sprint test, RS; Physical activity, PA.

no significant effect on children's physical activity. MQ was significantly affected by gender (F = 3.083, $p = 0.044$) and weight (F = 77.844, $p < 0.001$), but the interaction was not significant (F = 0.062, $p = 0.803$) (Table 1). As illustrated in Fig. 3, 64.12% of children in the OW/OB group had GMC at low or below levels, compared to 21.74% in the NW group.

In BCC, gender had a significant effect on static balance (F = 11.257, $p = 0.001$) and rhythm capacity (F = 29.611, $p < 0.001$), but did not significantly affect other sub-capacities ($p > 0.05$). Weight status had a significant effect on static balance (F = 9.3, $p = 0.003$) and dynamic balance (F = 11.164, $p = 0.001$). The interaction of gender and weight status significantly affected motor response (F = 1.471, $p = 0.024$) and kinesthetic differentiation (F = 5.454, $p = 0.02$), but did not significantly affect other sub-abilities ($p > 0.05$). The Fig. 4 showed indicators (motor reaction and kinesthetic differentiation) that are influenced by the interaction of weight status and gender.

Table 2 presented the results of the correlation analysis. BMI was significantly and inversely correlated with MQ (r = −0.543, $p < 0.001$), SLS (r = −0.282, $p < 0.001$), and YBT (r = −0.315, $p < 0.001$). MQ was significant positively correlated with SLS (r = −0.366, $p < 0.001$) and YBT (r = 0.402, $p < 0.001$), but negatively correlated with NMBR (r = −0.159, $p = 0.009$), TSBJ (r = −0.156, 0.011), and RS (r = −0.172, $p = 0.005$). No significant association was found between MQ and RT ($p > 0.05$). The was no significant correlation between activity and variables such as BMI, MQ, and BCC ($p > 0.05$).

Table 3 offered a comprehensive overview of the bootstrapping analysis. The analysis indicated that BMI serves as a significant predictor of MQ (β = −0.543, $p < 0.001$, 95%CI [1.773 to −1.214]). In a like manner, BMI emerges as a considerable determinant of YBT (β = −0.315, $p < 0.001$, 95%CI [−1.172 to −0.544]) and the SLS (β = −0.282, $p < 0.001$, 95% CI [−1.826 to −0.759]) respectively. However, the impact of BMI on other BCC's sub-abilities was not significant ($p > 0.05$).

Except for motor reaction ($p > 0.05$), other BCC significant predictors for MQ to varying degrees (YBT: β = 0.182, $p < 0.001$, 95%CI [0.082–0.284]; SLS: β = 0.189, $p < 0.001$,

**Table 3 Regression analysis of BMI, BCC and GMC.**

| | | β | SE | t | LLCI–ULCI | p | Adj.R² |
|---|---|---|---|---|---|---|---|
| X: BMI | Y: MQ | −0.543 | 0.142 | −10.518 | −1.773−−1.214 | <0.001 | 0.295 |
| | M1: YBT | −0.315 | 0.159 | −5.384 | −1.172−−0.544 | <0.001 | 0.99 |
| | M2: SLS | −0.282 | 0.271 | −4.772 | −1.826−−0.759 | <0.001 | 0.079 |
| | M3: NMBR | 0.024 | 0.025 | 0.394 | −0.040−−0.060 | 0.691 | 0.001 |
| | M4: TSBJ | 0.014 | 0.059 | 0.228 | −0.103−−0.130 | 0.820 | <0.001 |
| | M5: RT | −0.058 | 0.001 | −0.951 | −0.002−0.001 | 0.343 | 0.003 |
| | M6: RS | −0.074 | 0.009 | −1.202 | −0.29−0.007 | 0.231 | 0.005 |
| M1 | Y | 0.182 | 0.051 | 3.573 | 0.082−0.284 | <0.001 | 0.444 |
| M2 | | 0.189 | 0.030 | 3.797 | 0.054−0.172 | <0.001 | |
| M3 | | −0.117 | 0.320 | −2.443 | −1.412−−0.152 | 0.015 | |
| M4 | | −0.103 | 0.135 | −2.175 | −0.562−−0.028 | 0.031 | |
| M5 | | −0.012 | 9.853 | −0.243 | −21.799−17.007 | 0.808 | |
| M6 | | −0.157 | 0.905 | −3.248 | −4.719−−1.157 | 0.001 | |

Note:
Body mass index, BMI; motor quotient, MQ; Y-balance test, YBT; single leg standing test, SLS; Numbered medicine ball running test, NMBR; Target standing broad jump, TSBJ; Reaction time test, RT; Rhythmic sprint test, RS.

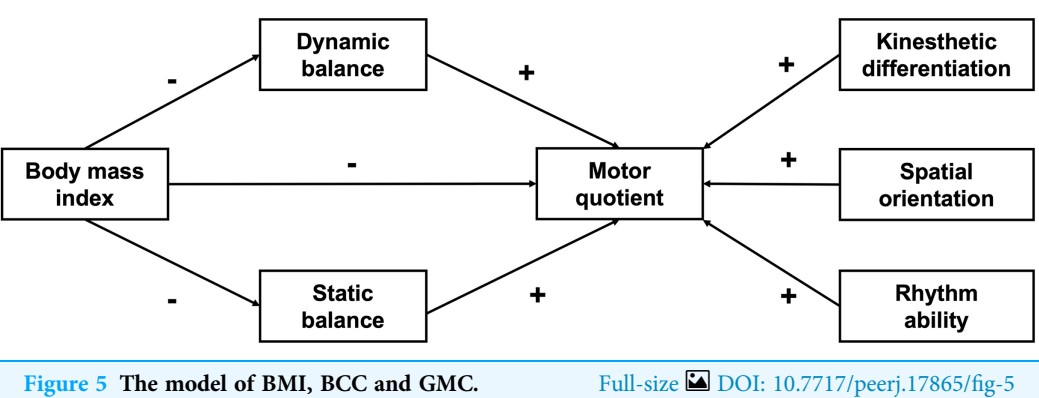

**Figure 5 The model of BMI, BCC and GMC.**

95%CI [0.054–0.172]; NMBR: $\beta = -0.117$, $p = 0.015$, 95%CI [−0.412 to −0.152]; TSBJ: $\beta = -0.103$, $p = 0.031$, 95%CI [−0.562 to 0.028]; RS: $\beta = -0.157$, $p = 0.001$, 95% [−4.719 to −1.157]), as shown in Fig. 5.

## DISCUSSIONS

This study found that overweight/obese (OW/OB) children in China have significantly poorer gross motor coordination (GMC), dynamic and static balance compared to their normal weight (NM) peers. In basic coordination capacities (BCC), We observed that all sub-abilities, except motor reaction, can positively influence GMC. However, only balance significantly relates to weight status, mediating the relationship between body mass index (BMI) and GMC.

### The effect of gender and weight status on GMC and BCC

The results indicated that Chinese children aged 9–10 who are OW/OB have significantly lower motor quotient (MQ), signifying poorer GMC. The results supported previous

studies demonstrating that excessive weight negatively impacts children's GMC (*D'Hondt et al., 2011*; *Kolic et al., 2020*). This association is attributed to a multitude of contributing factors. OW/OB children expend more energy in locomotor and basic stability. Additionally, they may be an anthropometric characteristic of long-term suboptimal lifestyle habits (*Nantel, Mathieu & Prince, 2011*). These children often show poorer cardiopulmonary function, decreased skeletal muscle strength/endurance, and lower levels of physical activity and sports experience. These factors, known to significantly affect GMC, have been extensively studied (*Lopes et al., 2011*; *Fransen et al., 2012*; *Barnett et al., 2016*; *Burton et al., 2023*). Therefore, OW/OB or higher BMI can impede both the long-term development and immediate performance of children's GMC through diverse mechanisms.

After converting MQ into categorical variables, we found that more than half of children might have suffered from varying degree of motor coordination disorders (MQ < 85). Overall, the mean MQ of all sample in this study was only 86.04, slightly higher than the findings of a 2022 study in China (85.65) (*Liu, Chen & Cai, 2022*). This further corroborated previous studies conducted in Europe, the United States, and other regions, which showed a stable and significant negative correlation between BMI/weight status and GMC. The influence of time and space on this relationship was very limited (*Hardy et al., 2012*; *Giuriato et al., 2021*). Notably, the study results showed that some NW children (21.74%) also had motor coordination problems. Without considering weight status, approximately half of the children (44.7%) had below normal GMC levels. Motor coordination problems seemed to be a common issue among children in global, increasingly appearing at young ages. Coupled with persistently high rates of OW/OB, this posed a serious threat to children's healthy development. Therefore, we suggested that there is an urgent need to establish appropriate strategies for improving GMC in young children.

Comparing the KTK sub-items score of the two groups, we found that the scores of OW/OB children were significantly worse across all items. This indicated that poor GMC in OW/OB children is not a coincidence and is not limited to specific motor skills, but more related to the individual's capacities. However, the degree of this difference depended on the characteristics of motor tasks, especially in walking backward (WB) and hopping for height (HH), where OW/OB children performed worst. This phenomenon may be related to the balance or stability. WB and HH require standing on a single leg and tandem (aligning both feet in a straight-line front and back), whereas jumping sideways (JS) and moving sideways test (MS) involve more common posture. Previous research indicated that there was no significant difference in static balance indicators between NW and obese groups when standing with their eyes open and bipedal (*Mi, Jingjing & Yang, 2021*). However, when standing on one foot, the balance performance of obese children was obvious worse ($F = 2.72$, $p < 0.05$). The reduction of the base of support increase the load on lower limbs, reduces proprioception, leads to sensory disturbances, and interferes stability, making the execution of gross motor skills more difficult (*Verbecque et al., 2021*). The results may also be affected by the primary direction of locomotion. Studies have demonstrated that OW/OB implies changes in the biomechanical characteristics of human

body, significantly altering movement patterns. For example, excessive accumulation of abdominal fat can shift the center of gravity forward, increasing the joint torque required, especially in the anterior-posterior direction (*Deforche et al., 2009*; *Alice et al., 2022*). An increase BMI reduced balance in the anterior reach direction but not in others (*Alhusaini, Melam & Buragadda, 2020*). The significant poorer performance of OW/OB children in Y-balance and single leg standing (assessing static and dynamic balance) also confirmed the aforementioned points. Hence, from the differential results, we can be inferred that the poorer GMC in OW/OB children appears to be due to weaker balance, which be further discussed below.

Both groups demonstrated that similar proficiency in BBC, except for balance, including spatial orientation, kinesthetic differentiation, motor reaction, and rhythm ability. This is supported by previous research, which did not find a significant correlation between body fat percentage (a more precise weight status predictor) and these capacities ($p > 0.05$). Instead, these capacities were associated with muscle mass ($p < 0.01$), total body water, and protein mass ($p < 0.05$) (*Willwéber & Čillík, 2017*). The study used BMI to assess the children's weight status, which may not be the most accurate measure. Research has shown that, unlike adults, annual increases in BMI during childhood are generally attributed to the lean rather than fat component of BMI (*Maynard et al., 2001*). In addition, the positive association between BCC and motor skills has been proven in several studies. The study on basketball players at different levels highlighted the significant contribution of spatial orientation in the coordination structure of the high-level group (12.16%), which was absent in the child and trainee-level groups. This is due to enhanced receptor sensitivity from lengthy basketball training (*Jerzy et al., 2015*). The results showed that the children's PA of different genders and weight statuses are similar. As a pioneer in the integration of physical education in China, Shanghai is highly representative nationwide. Most primary and secondary schools strictly follow the "Physical Education and Health Curriculum Standards (*Ministry of Education of the People's Republic of China, 2022*)" for objectives, content, and evaluation methods. Therefore, the curriculum structure and content across the three primary schools are very similar.

Research found that various associations between kinesthetic differentiation and specific sports, such as timing and accuracy of ball strikes in billiards and tennis (*Bańkosz, 2012*; *Polevoy, 2022*). The process by which living organisms execute motor actions involves perception, cognition, and action. In large-scale movements, such as gross motor skills, excessive weight primarily impacts the action levels. This increases the challenge of weight transfer and the motor output process from the nervous system to the muscles, rather than affecting perception and cognition. *Šimonek (2014)* refer to BCC as the "ability to coordinate information", suggesting that coordination involves identifying and integrating information. Although studies are showing a negative correlation between OW/OB and neurocognitive functions, including executive function, attention, and visual-spatial performance, this might only be relevant in the context of fine motor skill performance (*Liang et al., 2014*; *Mora-Gonzalez et al., 2019*), indicating that further is needed to explore the relationship between children's gross motor behaviors and neurocognitions.

Although OW/OB children tend to score lower on most tests compared to their NW peers, the influence of gender on coordination should not be overlooked. The unique changes that occur in girls during puberty lead to faster physical and mental development, resulting in significantly better performance in MQ and static balance compared to boys of the same age. The results indicate that girls have significantly better static balance. The gender difference in static balance ability among 9–10-year-old children has been confirmed in previous studies, with girls demonstrating stronger control over their center of gravity. This suggests that the integration system between the vestibular, visual, and proprioceptive senses, as well as small muscle responses, matures earlier in girls. In contrast, boys may not develop these abilities until around the age of 16 (*Kolic et al., 2020*). Additionally, the negative impact of being OW/OB on boys' motor response and kinesthetic differentiation abilities may be more pronounced than in girls. In the NW group, boys outperformed girls in both tests; however, there was a significant relative decline in the overweight/obese group. This phenomenon is related to the biological maturity of children. Research has found that girls enter and complete each stage of puberty earlier than boys, leading to more complete nervous system development at the same age. Consequently, girls have a stronger ability to perceive changes in the external environment, enabling more accurate adaptive adjustments in their movements (*Rogol, Clark & Roemmich, 2000*). These findings suggested that boys may need to pay more attention to their weight status during this period.

## The impact of BMI and BCC on GMC

This research also aimed to examine the impact of BMI and BCC on GMC. Initially, no significant correlation was found between motor reaction and MQ. This may be due to the testing method used. The KTK is designed to assess the lower limb and the whole-body motor coordination, and conducted in an environment that is relatively stable and predictable; whereas the motor reaction tests focused on coordination between visual input and upper limb motor output, emphasizing fine hand-eye coordination. Furthermore, the movement tasks in KTK seldom require participants to quickly to changes in the external environment. This suggests that future studies should explore the potential association between lower limb motor reaction and GMC.

Despite similar performance in kinesthetic differentiation, spatial orientation, and rhythm ability between NW and OW/OB children, these capacities all make a significant positive contribution to GMC. Kinesthetic differentiation is defined as a ability to precisely estimate the shape, distance, duration, and force needed for an action, based on sensory feedback (*Boichuk & Iermakov, 2020*). Previous studies have shown that kinesthetic differentiation is associated with maximum force level of left and right lower limb (r = 0.38 and 0.37) (*Harmaciński et al., 2016*). This capacity as a pivotal in enabling humans to adjust their behavior reasonably in unfamiliar movement environments (*Iorga et al., 2023*). The control of spatial orientation requires dynamic updates of the body-environment relational representation. It depends on the central integrating of current multisensory information and the contrasting sensory signals with planned trajectories, the body

schema, and previous experiences. Vision is the main sense in children's postural control strategies (*Mingxia & Qingwen, 2017*). However, relying on visual input can make it difficult to distinguish between self- and object-centered movement. Therefore, the vestibular and proprioceptive senses are crucial for identifying body position. The positive impact of spatial orientation on GMC highlights the significance of sensory redundancy in the motor performance of children in this age group. Rhythm capacity, an important element in motor control, learning, and development, involves grasp and replicating temporal beats and dynamic segmentation. A study by *Haines (2003)* showed a positive correlations between rhythm repetition and motor tasks and coordination in young children aged 4–8 years. Proficient rhythm capacity enables children to make rhythm posture adjustments in response to external environmental requirements or specific motor tasks. However, research by *Yin et al. (2023)* involving swimmers aged 8 to 12 indicated that rhythmic training can improve motor coordination, this enhancement effect but mainly in the JS (long jump) task. No significant changes were observed in the WB test before and after training (*Yin et al., 2023*). In conclusion, kinesthetic differentiation, spatial orientation, and rhythm capacity contribute uniquely to GMC, emphasizing different aspects of information selection and establishing a comprehensive cognitive framework that provides an objective and precise reference system for motion. Additionally, the combined functioning of these capacities underlines the crucial role of sensory system redundancy, a vital prerequisite for maintaining relative stability in children's GMC and minimizing the impact of external environmental disturbances.

These findings suggested that when executing motor tasks involving singular sub-capacity, performance is similar between NW and OW/OB children. Significant differences emerge in tasks requiring the integration of multiple capacities. The poorer GMC in OW/OB children may not be attribute to sensory disorders and deficiencies, but rather to difficulties in integration and cognition.

Although the aforementioned sub-capacities are no significantly associated with BMI, balance showed correlation with both BMI and GMC. Therefore, this study further explored the mediating role of balance between BMI and GMC. The result indicated that BMI can negatively impact on GMC both directly and in directly through balance, with static and dynamic balance serving as a parallel mediator between BMI and GMC (Fig. 5). When facing the same balance disturbances, OW/OB children need to employ more advanced motor strategies to maintain or restore basic stability. Balance reflects the level of muscle synergy in children in some extent, but children with coordination disorders need to use more hip and ankle strategies, implying activation of more muscle groups (*Fong et al., 2015*). Unfortunately, this research did not consider weight status. This suggested that OW/OB need to utilize additional resources at musculoskeletal level to maintain stability. In psychology, a study based on the "limited attentional resources" hypothesis using dual-task paradigm showed that children performed similar in isolated balance tests, when one group needed to perform additional tasks (such us computing and writing) while maintaining balance, the quality of execution and balance scores both significant decreased. The poorer GMC in OW/OB children may be due to the same intrinsic

mechanism linked to balance. Their excess weight does not create ideal conditions for postural stability, they need to allocate resources to maintain or recover balance, such as strength, attention, and perception, which are important relevant factors of GMC (*Jelsma et al., 2021*).

In summary, this study initially constructed a model of BMI and BCC affecting children's GMC. The results suggested that static and dynamic balance play parallel mediating roles between BMI and GMC. Despite not significant related to BMI, spatial orientation, kinesthetic differentiation, and rhythm ability make positive contributions to GMC. Interestingly, GMC does not appear to have a significantly associated with the motor reaction. Further research is required to confirm the validity and generalizability of our findings. And, refining the indicators related to GMC by integrating various aspects such us biomechanics, biology and psychology, to lay the foundation for establishing a comprehensive model.

The strength of our study lay in its preliminary exploration of the association between two concepts of motor coordination in practice, based on "child OW/OB and GMC problems", which provided a new perspective for future research: the necessity to discuss various forms of motor coordination. To my knowledge, there has been little research that simultaneously investigated the relationship among different coordination. However, this study had several limitations. The absence of tests specifically designed for BCC in school-aged children may mean that the results not accurately reflect their true levels. Future research should aim to develop and establish a more appropriate assessment system. Additionally, due to time and economic constraints, this study did not include anthropometric indices other than BMI to fully assess the children's weight status. The cross-sectional design limited our ability to establish causal relationships between weight status, GMC and BCC. We plan to conduct a longitudinal study in the future.

## PRACTICAL IMPLICATION

This study emphasized the important of individual's capacities for GMC, providing a basis for developing exercise strategies. We recommend that coaches and teachers incorporate elements of these sub-capacities into strategy formulation and implementation, as this is more likely to yield favorable training outcomes. Additionally, when assessing GMC, special attention should be given to children's stability and balance performance.

## CONCLUSION

This cross-sectional study provided initial insights into the potential relationships between children's BMI, PA, gender, sub-capacities of BCC, and GMC. Excluding motor reaction, the other sub-capacities, including kinesthetic differentiation, spatial orientation, balance, and rhythm capacity, contribute to GMC at different degrees. It is noteworthy that both static and dynamic balance are affected by BMI, which is closely associated with reduced GMC in OW/OB children compared to their NW peers.

### Funding

This work was supported by the Shanghai Key Lab of Human Performance (Shanghai University of Sport) (No. 11DZ2261100). The funders had no role in study design, data collection and analysis, decision to publish, or preparation of the manuscript.

### Grant Disclosures

The following grant information was disclosed by the authors:
Shanghai Key Lab of Human Performance (Shanghai University of Sport): 11DZ2261100.

### Competing Interests

The authors declare that there are no competing interests.

### Author Contributions

- Yuan Sui conceived and designed the experiments, performed the experiments, analyzed the data, prepared figures and/or tables, authored or reviewed drafts of the article, and approved the final draft.
- Lin Cui conceived and designed the experiments, prepared figures and/or tables, and approved the final draft.
- Binbin Jia conceived and designed the experiments, analyzed the data, prepared figures and/or tables, authored or reviewed drafts of the article, and approved the final draft.
- Xiangyang Ding conceived and designed the experiments, performed the experiments, authored or reviewed drafts of the article, and approved the final draft.
- Min He conceived and designed the experiments, performed the experiments, authored or reviewed drafts of the article, and approved the final draft.
- Yingen Da conceived and designed the experiments, performed the experiments, authored or reviewed drafts of the article, and approved the final draft.
- Yue Shi conceived and designed the experiments, analyzed the data, prepared figures and/or tables, authored or reviewed drafts of the article, and approved the final draft.
- Fei Li conceived and designed the experiments, performed the experiments, analyzed the data, prepared figures and/or tables, authored or reviewed drafts of the article, and approved the final draft.
- Pan Li conceived and designed the experiments, prepared figures and/or tables, authored or reviewed drafts of the article, and approved the final draft.

### Human Ethics

The following information was supplied relating to ethical approvals (*i.e.*, approving body and any reference numbers):

Shanghai University of Sport

### Data Availability

The raw data is available in the Supplemental File.

## Supplemental Information

Supplemental information for this article can be found online at http://dx.doi.org/10.7717/peerj.17865#supplemental-information.

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
