# Peer review of "Correlation between gross motor coordination and basic coordination capacities in normal-weight and overweight/obese children aged 9–10 years"

_PeerJ, doi:10.7717/peerj.17865_

## Round 0.1 · original submission · Minor Revisions

Both reviewers made minor comments that will help to improve the manuscript. Please address those and resubmit. Some comments ask for further information which I assume you do not have since it would have been included if you did. For example the request for more information about other sports some children may have participated in. So, you do not need to add the data, just explain why it is not included in your response to the reviewers.

Reviewer 1 ·

Basic reporting

The manuscript is well structured, the sections are presented logically, the ideas expressed are clear and coherent. The references are well selected and associated with the studied topic. The introduction provides a good analysis of the independent, mediating and dependent concepts/variables analyzed in the research (BMI, BCC and GMC/MQ). The discussion section presents similar investigations and reports its own results to these investigations. The results and graphs presented summarize the most relevant data/statistical results of the investigation.

Experimental design

The two research directions are correctly formulated at the end of the introduction. The description of the associated GMC and BCC tests provides important details regarding the execution technique, the routes taken and the quantification/measurement of the results. The documents associated with the manuscript confirm compliance with the ethical standards required for scientific research involving human subjects.

Validity of the findings

The investigation provides information that complements the topic researched in the scientific literature, related to the relationship between BMI values and elements of coordinative capacity, for children in the first years of puberty. The applied statistical procedures allow the analysis of the correlations between the variables and the differences between the NW sample vs. OW/OB for the investigated parameters. The conclusions summarize the research results very well and are correctly associated with the objectives of the study.

Additional comments

1. It would be useful to provide additional information related to the level of involvement of students in physical activities. Are there also children who practice the different sports? This aspect would strongly and favorably influence the results of the coordination tests, but also the BMI values. Are the curriculum activities/contents and physical exercises in PE lessons identical for the 3 schools in Shanghai?
2. Rhythm ability test and Spatial orientation test (components of BCC): Were students allowed to practice before assessment on these tests to memorize/understand the specific routes and requirements? Is there a value scale for quantifying performance (by age category) for these two coordination tests?
3. Maybe you can specify in how many assessment sessions you applied the tests and what the body warm-up consisted of before the assessment.
4. A gender performance comparison (male vs. female) would be useful, which you could present in other studies. It is possible that puberty-specific transformations influence the results of coordination tests, especially for balance (maturation is faster for girls).
5. Lines 234-238 / The results shown in the comparison of mean values between batches (t and Sig.) could be transferred to table 1 (where you have the mean values included), I think it would be easier to read that way.
6. Lines 399-400 / The findings suggested that BMI could have a negative effect on MQ both directly and indirectly through static and dynamic balance (figure 2). I'm thinking that Table 2 (where you included the Pearson correlation coefficient values) is correct. Figure 2 is Test diagram of spatial orientation.

---

## Round 0.2 · accepted · Accept

Thank you Dr Sui et al for comprehensively addressing the concerns and suggestions of the reviewers. I have checked the tracked manuscript and response t reviewers and believe that the manuscript is now ready for publication.